# Electrostatic Precipitators as an Indoor Air Cleaner—A Literature Review

**Alireza Afshari** [1,*]**, Lars Ekberg** [2]**, Luboš Forejt** [3]**, Jinhan Mo** [4,5,*]**, Siamak Rahimi** [1]**, Jeffrey Siegel** [6]**, Wenhao Chen** [7]**, Pawel Wargocki** [8]**, Sultan Zurami** [9] **and Jianshun Zhang** [10,11,12]

1 Department of the Built Environment, Aalborg University, A.C. Meyers Vaenge 15, DK-2450 Copenhagen, Denmark; siamak.ardkapan@airlabs.com
2 CIT Energy Management AB, SE-412 88 Gothenburg, Sweden; lars.ekberg@cit.chalmers.se
3 Honeywell Aerospace, 78365 Hlubočky-Mariánské Údolí, Czech Republic; lubos_48y@yahoo.com
4 Department of Building Science, Tsinghua University, Beijing 100084, China
5 Beijing Key Laboratory of Indoor Air Quality Evaluation and Control, Beijing 100084, China
6 Department of Civil & Mineral Engineering, University of Toronto, Toronto, ON M5S 1A4, Canada; jeffrey.siegel@utoronto.ca
7 Indoor Air Quality Program, Environmental Health Laboratory Branch, California Department of Public Health, Richmond, CA 94804, USA; wenhao.chen@cdph.ca.gov
8 International Centre for Indoor Environment and Energy DTU Civil Engineering, Technical University of Denmark, 2800 Kongens Lyngby, Denmark; paw@byg.dtu.dk
9 CREATE Campus for Research Excellence and Technological Enterprise, Singapore 13860, Singapore; zee-ms@hotmail.com
10 Building Energy & Environmental Systems Laboratory (BEESL), Department of Mechanical and Aerospace Engineering, Syracuse University, New York, NY 13244, USA; jszhang@syr.edu
11 Department of Civil and Environmental Engineering, Syracuse University, New York, NY 13244, USA
12 School of Architecture and Urban Planning, Nanjing University, Nanjing 210093, China
* Correspondence: aaf@build.aau.dk (A.A.); mojinhan@tsinghua.edu.cn (J.M.)

**Abstract:** Many people spend most of their time in an indoor environment. A positive relationship exists between indoor environmental quality and the health, wellbeing, and productivity of occupants in buildings. The indoor environment is affected by pollutants, such as gases and particles. Pollutants can be removed from the indoor environment in various ways. Air-cleaning devices are commonly marketed as benefiting the removal of air pollutants and, consequently, improving indoor air quality. Depending on the type of cleaning technology, air cleaners may generate undesired and toxic byproducts. Different air filtration technologies, such as electrostatic precipitators (ESPs) have been introduced to the market. The ESP has been used in buildings because it can remove particles while only causing low pressure drops. Moreover, ESPs can be either in-duct or standalone units. This review aims to provide an overview of ESP use, methods for testing this product, the performance of existing ESPs concerning removing pollutants and their byproducts, and the existing market for ESPs.

**Keywords:** Electrostatic precipitator (ESP); indoor air quality; filter; filtration; air pollutions

---

## 1. Introduction

Many people spend most of their working and living hours in an indoor environment [1]. The indoor environmental quality (IEQ) is important for human health, wellbeing, and productivity at work. IEQ is affected by a number of factors, among others, gases and particles. Airborne particles vary in size, form, and chemical composition. Their size ranges from a few nanometers up to tens of micrometers. It has been established that fine and ultrafine particles may have adverse health effects on the human body [2,3].

Particles are important indoor pollutants. The increase in particle concentration is associated with airway inflammations and reduced lung function [4]. A recent study on the effect of long-term exposure to traffic particles confirms that particles cause a decline in the lung function of elderly people [5]. A study of the acute effect of particles in China reveals a direct association between the number of emergency room visits and particle concentration [6]. In addition, studies have focused on finding out how particles contribute to cardiovascular diseases and mortality [7–10].

There are various ways in which pollutants can be removed from the indoor environment. If the outdoor particle level is low, ventilation reduces the concentration of indoor particles by means of dilution. If the outdoor particle level is high, ventilation will increase indoor particle pollution and the particles need to be removed by filters of the supply air or in-room standalone air cleaners. One practical application is to investigate the possibility of using air recirculation together with air cleaners as a technical solution to improve indoor air quality, while reducing the outdoor air supply and hence, energy used for ventilation and air conditioning, including air cleaning. The energy used for air conditioning of buildings makes up almost 40% of the total building energy consumption [11].

Air cleaning devices are commonly marketed as benefitting the removal of air pollutants and consequently, improving indoor air quality [12]. Depending on the type of cleaning technology, air cleaners may generate undesired and toxic by-products and contribute to secondary emissions such as ozone and aldehyde, and their effectiveness may vary [13,14].

Different air filtration technologies have been introduced to the market such as mechanical filters, corona dischargers, and electrostatic precipitators [14]. The electrostatic precipitator (ESP) has been used as an air cleaning technology in mechanical ventilation systems in residential buildings, since it can remove particles while only causing low pressure drops. Electrostatic precipitators can be either in-duct or stand-alone units.

The main modus operandi of ESPs is the use of a high voltage power supply to establish a strong electric field to charge particles in the air and then, collect the charged pollutants at a later stage by an oppositely charged plate. Therefore, the particles migrate rapidly to the collection surface due to the charges [15]. This high level of voltage may cause some other reactions such as ozone generation. However, it should be noted that smaller particles have higher mobility and are more easily attracted by lower charge levels. In addition, the electrostatic deposition velocity of a small particle is higher than diffusion velocities and gravitation velocities. Electrostatic precipitators can offer some benefits over other highly effective air filtration technologies. For example, the high-efficiency particulate air (HEPA) filtration requires filters and may become "sinks" for some harmful forms of bacteria as well as causes high pressure drops. Therefore, a thorough examination of the performance of ESPs is required to obtain a complete overview of this technology. An experimental study shows that a novel ESP that uses anticorrosive materials can generate a large number of unipolar ions whilst producing only a negligible concentration of ozone, and achieve a strong collection performance of more than 95% for ultrafine particles, while only consuming a power of 5 W and generating a pressure drop of 5 Pa per 1200 $m^3$/h [16].

Some procedures were introduced to examine the performance of air cleaning technologies, including electrostatic precipitation, and some standards (e.g., ANSI/AHAM AC-1 [17]; GB/T 18,801 [18]) were established to test the pollutant removal ability of the air cleaners placed in a duct. The existing procedures and standards and the existing market for ESPs [19] are discussed in detail in this review.

This review aims to give an overview of ESP use, methods for testing this product, the performance of existing ESPs in removing pollutants, their by-products, and the existing market for ESPs.

## 2. Operation of Electrostatic Precipitators

### 2.1. Principle of ESPs

Electrostatic precipitation uses the forces of an electric field on charged particles to separate particles from a gas stream. The particle is deliberately charged and passed through an electric field,

causing the particles to migrate towards an oppositely charged electrode that acts as a collection surface. Commercial ESPs accomplish charging by using a high-voltage, direct-current corona surrounding a highly charged electrode, such as a wire. The large potential gradient near the electrode causes a corona discharge comprising electrons. The gas molecules become ionized with charges of the same polarity as the wire electrode. These ions then collide with and attach to the aerosol particles, thereby charging the particles. The basic processes in an electrostatic precipitator are shown in Figure 1.

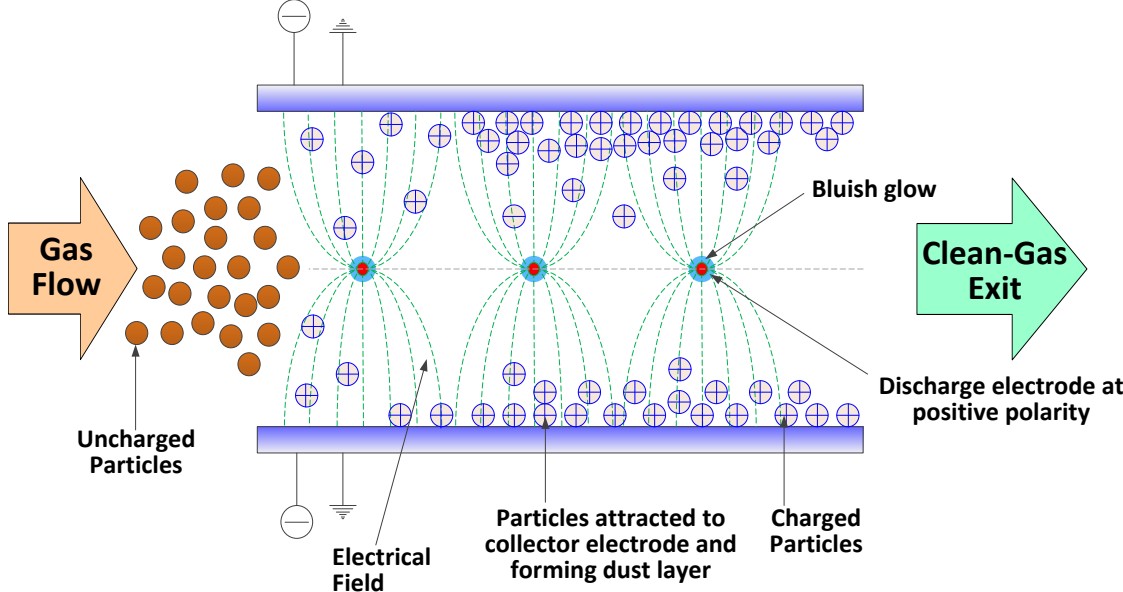

**Figure 1.** Schematic of the basic processes of an electrostatic precipitator (Source: modified from a guide document published by Ohio Environmental Protection Agency, USA, accessible at: www.epa.state.oh.us/portals/27/engineer/eguides/electro.pdf).

### 2.2. Corona Generation

When the voltage difference between the wire and plate electrodes increases, an electrical breakdown of the gas occurs near the wire. When gas molecules get excited, one or more of the electrons can shift to a higher energy level. This state is transient; once the excitation has ceased, the molecule reverts to its ground state, thereby releasing energy. Part of this energy converts to light. The bluish glow adjacent to the wire is the corona discharge, as shown in Figure 1.

The space between the wire and the plate can be divided into an active and a passive zone (see Figure 2). In the active zone, defined by the corona glow discharge, electrons leave the wire electrode and impact gas molecules, thereby ionizing the molecules. The additional free electrons also accelerate and ionize more gas molecules. This avalanche process continues until the electric field decreases to the point when the released electrons do not acquire sufficient energy for ionization [20]. The radius of the active zone is roughly 0.197 mm if the wire radius is 0.1 mm [21], and the length of the passive zone varies in different studies [22].

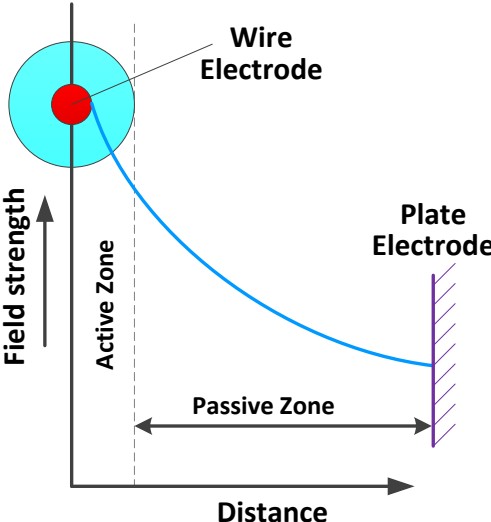

**Figure 2.** Variation of field strength between wire and plate electrodes [20].

A negative corona is formed if the discharge electrode is negative and a positive corona is formed if the discharge electrode is positive. In a negative corona, positive ions are attracted toward the negative wire electrode, and electrons are attracted toward the positive plate. Beyond the corona glow region, the electric field diminishes rapidly, and if electronegative gases are present, the gas molecules become ionized by electron impact. The negative ions move toward the plate electrode. In the passive zone, these ions attach themselves to aerosol particles. When a corona is negative, the free electrons leaving the active zone on their way to the plates are transformed into negative ions with substantially lower mobility. The negative charge carriers thus cover the first part of their trajectory as fast, free electrons, and the second part as slower ions; their average mobility is lower than that of free electrons but higher than that of the large ions. Thus, a negative corona is manifested in a non-uniform corona. On the other hand, when a corona is positive, the positive charge carriers are large, slow ions by the origin and retain this form throughout their motion. Therefore, a positive corona is manifested as a uniform field between wire and plate. Consequently, a negative corona always has a higher corona current than a positive corona for an applied voltage. In contrast, a positive corona has a much lower density of free electrons compared to a negative corona [23].

Nevertheless, a negative corona generates much more ozone than the corresponding positive corona. As the reactions that produce ozone are relatively low energy, the greater number of electrons of a negative corona leads to increased production of ozone. Therefore, a positive corona is usually used for cleaning the air in occupied spaces.

*2.3. Types of ESP*

ESPs can be classified according to a number of features in their design, such as the structural design and operation of the discharge electrodes (rigid-frame, wires, or plate) and collection electrodes (tubular or plate). A common method of classifying ESPs is by the number of stages used for charging and removing particles from a gas stream.

2.3.1. Single-Stage

When the same set of electrodes is used for both charging and collecting, the precipitator is called a single-stage precipitator. Single-stage ESPs use very high voltage (50 to 70 kV) to charge particles [24]. After being charged (by ionized air molecules), particles move in a direction perpendicular to the gas flow through the ESP and migrate to an oppositely charged collection surface, usually a plate or tube. Figure 1 shows a typical single-stage wire plate precipitator with positive corona discharge. Single-stage ESPs are usually used in industrial applications where dust loadings are higher [20].

### 2.3.2. Two-Stage

If different sets of electrodes are used for charging and collecting, the precipitator is called a two-stage precipitator (see Figure 3). The charging field and the collecting field are independent of each other. In a two-stage ESP, the charging stage, which is located upstream of the collection stage, consists of a series of small, positively charged wires equally spaced at 2.5 to 5.1 cm (1 to 2 in.) from parallel grounded plates. A corona discharge between each wire and a corresponding plate charges the particles suspended in the airflow as they pass through the charging stage. The direct current voltage applied to the wires is approximately 12 to 13 kV [24]. The charging stage is short, providing a short residence time, and the collection stage is five or more times longer to provide sufficient time for collection [25].

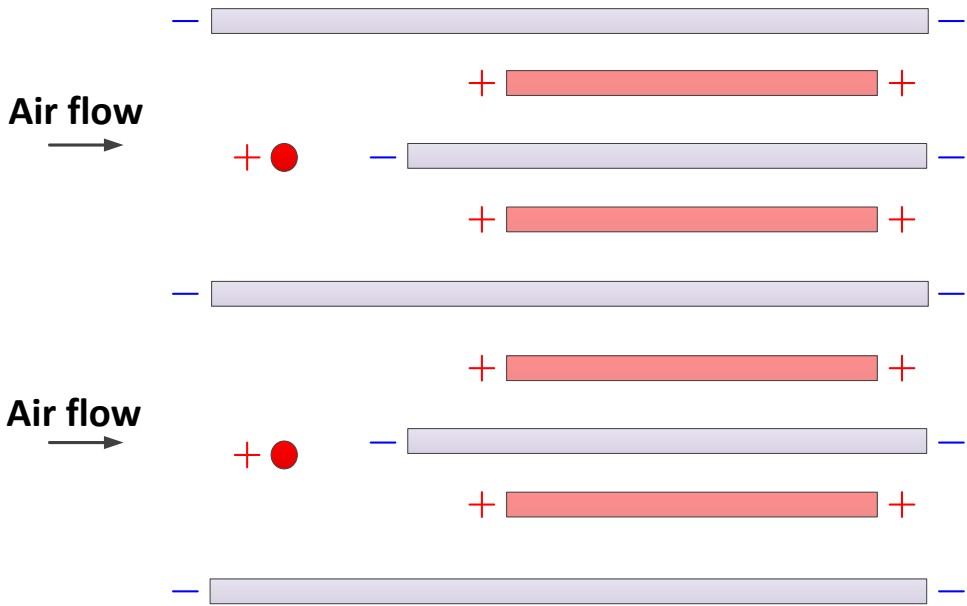

**Figure 3.** Conceptual diagram of two-stage positive corona electrostatic precipitators [26].

## 3. Testing and Standards

Currently, testing and standards assessing indoor air associated with the performance of electrostatic precipitators relate to those used as portable air cleaning units for room-size applications or those meant to be installed in the heating, ventilation, and air conditioning HVAC system for whole building applications. Regardless of the application, these standards assess electrostatic precipitators' performance by either evaluating their removal of particles from the air or determining their ozone generation as by-product. These standards are created with the intent to protect human occupants by reducing harmful exposures to airborne particles and ozone in controlled chambers or test rig environments. The standards and procedures are described in greater detail below.

### 3.1. Standards and Procedures for ESPs in Portable Air cleaning Units (PACs)

### 3.1.1. Particle Removal

The pull-down test method is the most commonly used in standards or protocols for assessing the removal of particles in the air using PACs. This method is also applicable to ESPs. Standards or protocols using this method include the ANSI/AHAM AC-1 standard [17], NRC (National Research Council Canada) protocol [27], NCEBT (National Center for Energy Management and Building Technologies) method [28], China standard [18], and the Swiss standard [29]. The pull-down test method is not exclusively used on PACs with ESP technology as it can be applied to other technologies (e.g., media filtration, photocatalytic) as well. The pull-down test method typically involves particles

being dosed into a chamber containing the PAC to be tested and observing first-order decay of particle concentrations with and without the PAC in operation. The difference in particle decay is used to determine the performance of the PAC.

Standards or protocols differ in terms of particles being used as challenge aerosols as well as an index to characterize PAC performance. In the former, challenge aerosols used may give consumers information on the PAC performance in removing certain types of particles. It is also noteworthy that the challenge aerosols can provide information on the PAC performance in removing particles of different sizes. Considering that ESP technology has been promoted as being efficient for the removal of ultrafine particle (UFP), only a few standards consider UFP removal performance. In terms of performance index, the most commonly used is the device clean air delivery rate (CADR) values measured in cubic feet per minute (cfm) or cubic meter per hour (cmh). Depending on the challenge particles, CADR values are reported for the removal of particle types or particle sizes. The Swiss, Chinese, and Japanese standards used the concept of half-life to report the performance of PAC for particle removal. The AHAM, China, and NRC standards relate the CADR performance obtained in chamber settings to actual service conditions by recommending room sizes to achieve an 80% indoor particle concentration reduction under steady-state conditions. The NRC protocol developed a MERV (minimum efficiency reporting value)-like particle removal rating to rate PACs. Details of particle challenges and performance index differences are summarized in Table 1.

**Table 1.** Standards and procedures for evaluating the initial performance of electrostatic precipitators (ESPs) in portable air cleaning (PAC) and in-duct systems.

| Standard/Protocol (Ref.) | Country | Method | Challenge Particles | Measured Particle Size Range | Performance Index |
|---|---|---|---|---|---|
| | | | Portable Air Cleaners | | |
| ANSI/AHAM [17] | US | Pull-down | Environmental Tobacco Smoke Arizona Road Dust Paper Mulberry Pollen | 0.1 to 1.0 μm 0.5 to 3.0 μm 5 to 11 μm | CADR [a] |
| GB/T-18801 [18] | China | Pull-down | Environmental Tobacco Smoke Arizona Road Dust Paper Mulberry Pollen | 0.1 to 1.0 μm 0.5 to 3.0 μm 5 to 11 μm | CADR |
| NRC Protocol [27] | Canada | Pull-down | Polydisperse Potassium chloride (KCl) | 50 nm to 5 μm | CADR |
| NCEMBT Procedure [28] | US | Pull-down | Polydisperse potassium chloride (KCl) | 0.1 to 11.5 μm | CADR |
| Lucerne University (2012) [29] | Switzerland | Pull-down | ISO 12103-1 A1 Ultrafine test dust. | 0.2 to 5 μm | |
| JIS C 9615 [30] | Japan | Single-pass | JIS Z 8901 standard dusts | . . . | Removal rate |
| XP B44-200 [31] | France | Single-pass | DEHS, cat allergens, *Staphylococcus epidermidis Aspergillus niger* | 0.3 and 5 μm | SPE [b], CADR |
| | | | In-duct air cleaners | | |
| ANSI/AHRI 681 [32] | US | Single-pass | Polydisperse potassium chloride (KCl) | 0.3 μm to 10 μm | SPE |

Notes: (a) CADR: clean air delivery rate; (b) SPE: single-pass efficiency.

The other method for assessing ESPs in PAC performance is the single-pass efficiency test method, which is an approach similar to the ASHRAE standard 52.2 method for testing media filters in a test rig. The French standard, XP B44-200 [31], measures upstream and downstream concentrations of Di-Ethyl-Hexyl-Sebacat (DEHS) (between 0.3 and 5 μm) particles, cat allergens, *Staphylococcus epidermidis*, and *Aspergillus niger* in a special chamber for PACs. The removal efficiencies and CADR of the particles are given. Although the pull-down test method and single-pass efficiency method are theoretically related, air mixing, PAC, and/or chamber short-circuiting may violate the relationship [33]. The Japanese standard also employs a single-pass test using a special chamber [30]. Upstream and downstream filter light transmittances are used to evaluate the removal rates of standardized challenge particles.

The AHAM AC-3 standard [34], JIS 9615 standard [30], China standard [18], and the Swiss procedure [29] are the only published standards available which evaluate long-term PAC particle removal performance. In these standards, known amounts of particles are artificially loaded onto PACs in a chamber dedicated to simulating long-term operation under a "standard" condition. Upon loading, the PACs are subjected to an initial performance evaluation [17]. The JIS standard includes the particle capacity test by determining total particle amounts following an 80% flow reduction or if the removal rate decreased by 85%.

### 3.1.2. Ozone Production

There are two methods for assessing ozone production from portable ESPs in PACs: (1) concentration measurement; and (2) generation rate determination. For the concentration measurement method, an ozone production test standard procedure has been included in the US Underwriters Laboratory (UL) standard 867 [35]. According to the UL standard, the ozone concentration should not exceed 0.05 ppm after 24 h of continuous operation of a cleaner in an enclosed chamber of 31.1 m$^3$, and the interior surface must be made of stainless steel or other nonporous and nonreactive material. The UL standard 867 specifies that the ozone must be measured at 50 mm downstream of the product air outlet, which is primarily a measure of the outlet concentration instead of the chamber concentration. As a result, the actual ozone generation rate of the air cleaner and its influence on the room ozone concentration depends on the airflow rate of the air cleaner. In addition, the size of a typical bedroom can be smaller or larger than the size specified, and the actual indoor surface materials can be different from those in the UL standard test chamber. It may be a concern that an ESP-based air cleaner that has passed the UL standard test may still pose an ozone exposure hazard to occupants because of differences in room sizes and deposition velocities associated with different interior surfaces.

According to the CSA C-187 Cl. 7.4 [36] standard, the 8-hour time-weighted average (TWA) ozone concentration from ESPs measured for 24 h should not exceed 0.05 ppm, and was updated to be 0.02 ppm in 2016. This standard requires measurements in a chamber similar in size to that of the standard UL 867, but performed under static conditions.

Other standards or procedures have been proposed with a method to calculate the ozone generation rate, which is an intrinsic property of ESP. The methods involved using a well-mixed and positive pressured chamber supplied with air filtered for particles and ozone. Typically, two tests are made for measuring ozone generations, first with PAC powered on and the second with PAC powered off to obtain deposition loss to surfaces. From the measured data, the ozone generation rate of the PAC is calculated and then modelled to determine the predicted indoor ozone concentration in actual buildings. The NCEMBT procedure [28] calculates ozone generation rates (in mg/hr) of PACs using the measured ozone concentration in a 55 m$^3$ stainless steel chamber, but does not provide guidance from the PAC on expected ozone concentrations in actual residences. The NRC protocol [27] measures PAC ozone generation rates from steady-state ozone concentration in a chamber similar to that used in the NCEMBT procedure and also suggests that PACs may or may not exceed the indoor ozone concentration set by the Health Canada guideline of 50ppb based on a "typical" Canadian residential bedroom.

*3.2. Standards and Procedures for ESPs in In-Duct Systems*

3.2.1. Particle Removal

The ASHRAE Standard 52.2 for evaluating filter performance in in-duct systems is not applicable for ESPs [37]. Currently, only one standard evaluates ESP performance in in-duct systems. The ANSI/AHRI Standard 681 (only applicable to residential conditions) [32] measures the performance of particle removal of ESPs under Group RII (electronic air cleaner). The test apparatus requirements and qualification followed that of the ASHRAE Standard 52.2. Standard 681 evaluates the size efficiency of ESP particle removal for 12 particle size ranges or bins between 0.30 and 10 μm and the ESP dust holding capacity. A polydisperse solid-phase potassium chloride (KCl) particle generated from an aqueous solution is used in this standard to determine particle size efficiency.

3.2.2. Ozone Generation

For ozone generation evaluation from ESPs in HVAC systems, the method used in current standards and procedures only measures ozone concentration and not generation rates. The CSA C-187 [36] standard measures an 8-hour time-weighted average (TWA) ozone concentration from ESPs. The standard requires that the ozone concentrations do not exceed Health Canada's 50 ppb indoor guideline for ozone. The ANSI/AHRI Standard 681 [32] also requires electronic air cleaners to be tested for ozone concentration at the maximum rated airflow rate, as published by the manufacturer. The standard requires that ESPs should have a maximum ozone concentration in the effluent air not exceeding 50 ppb. Both these standards measure the ozone concentration using the ASHRAE 52.2 test rig and rely on specific flow rate requirements to obtain downstream ozone concentrations.

A recent standard test method calculates in-duct ESP ozone mass generation rates using a modified test rig similar to that of ASHRAE 52.1 and EN779 [38]. This standard test determines the ozone mass generation rate as the product of the average ozone mass concentration increase across the ESPs and its volumetric flow rate.

In accordance with California Assembly Bill 2276 (2006, Pavley), the California Air Resources Board (CARB) adopted an air cleaner regulation to limit the amount of ozone produced from indoor air cleaning devices [39]. The regulation became final on 18 October 2008. Additionally, several amendments to the regulation received final state approval on 10 September 2010.

## 4. Measurements of Particle Removal in Field and Laboratory

Air cleaners were tested in laboratory settings to ensure that the equipment meets certain quality criteria with respect to air cleaning performance, and to ensure that they do not produce harmful substances. However, there are gaps between the laboratory test procedures and the use of the equipment in "real-life" situations. This section intends to illustrate some of these gaps.

*Electrostatic Precipitator vs. Mechanical Filtration*

Noise: Chen et al. [28] reported the results of tests of six different portable air cleaners based on various filtration technologies. Of the tested units, four had sorption filters in combination with mechanical particle filters, and one of the units had a sorption filter in combination with an electrostatic precipitator. The latter unit showed noise generation towards the lower end of the sound pressure levels observed for units equipped with a fan. At maximum capacity, the sound pressure level measured in an acoustic chamber was 50 dB(A). Of the units with a mechanical filter, one showed 47 dB(A), while three units showed 57 dB(A) or higher. Thus, the results did not indicate any clear difference between the electrostatic air cleaner and the units with mechanical filters, as regards noise generation. Furthermore, Zuraimi [27] concluded that, in terms of noise generation, electrostatic precipitators are comparable to media-based portable air cleaners.

Power: Portable air cleaners are equipped with an integrated fan, which in some cases, can be controlled. One reason to control the capacity of an air cleaner may be to reduce the use of electricity

and another to reduce noise. Thus, the occupants may switch the unit to a lower capacity if the noise due to the air cleaner operation is perceived to be unacceptable.

CADR: As reported by Zuraimi [27], the electrical power need of electrostatic precipitators may be comparable to that of units equipped with mechanical filters. However, there is a reason to consider particle removal capacity when evaluating the need for electricity. The electrostatic precipitator tested by Chen et al. [28] showed the highest particle removal efficiency and the highest CADR of all tested units. Thus, the specific electrical power, normalized with respect to CADR, was 0.2 W per cfm (CADR) for the electrostatic air cleaner, while the other units typically needed far more than twice the power (i.e., 0.4–1.4 W/cfm CADR).

Mølgaard et al. [40] conducted tests on five portable air cleaning units, one of which was an electrostatic precipitator. The conclusion was that the tested filter-based units showed better performance than the tested electrostatic unit did. However, this conclusion should not be regarded as a general difference between filter-based air cleaners and electrostatic air cleaners. Tian et al. [41–44] developed electrostatically assisted mechanical filters. Tian et al. [45] also presented a comprehensive quality factor (*CQF*) to evaluate the performance of electrostatic particle removal technologies.

$$CQF = \frac{-\ln(1-\eta)}{\Delta p + \Delta p'} = \frac{-\ln(1-\eta)}{\Delta p + \frac{\eta_{fan}P}{u_0}} \tag{1}$$

where $\eta$ and $\Delta p$ are the single-pass filtration efficiency and pressure drop (Pa) of the filter; $\Delta p'$ is the apparent pressure drop (Pa), which is equivalent to that of a normal mechanical filter with a specific fan energy consumption of $P$ (W/m$^2$); $\eta_{fan}$ is the energy efficiency of the driven fan; $u_0$ is the face velocity (m/s).

In summary, it is not possible to make firm generalizations in favor of one technology compared to another, i.e., electrostatic vs. mechanical filtration. Particle removal capacity, noise, energy consumption, and by-product emission are four important aspects that need careful consideration when evaluating portable air cleaners.

## 5. By-Product and Secondary Emissions

Hazardous by-product generation has been a concern for the use of electrostatic precipitators (ESPs) in indoor as well as transportation environments. This section summarizes the major literature findings on the generation mechanism, influencing factors, range of generation rates, and its impact on indoor ozone exposure for by-products from ESPs.

### 5.1. Ozone

It is widely known that ozone can be generated from corona discharge and/or the ionization process [46,47]. Both ESPs which have a fan and collection plates, and smaller ion generators, which often do not have a fan and may or may not have collection plates, are ionizers. They charge incoming particles with a corona and may, therefore, produce ozone [33]. Although the focus of this review paper is the performance of ESPs, the literature reporting ozone generation sometimes contained both ESPs and small ion generators in tests and by examining the literature, we cannot identify whether those that generated ozone were ESPs. Therefore, the ozone generation rates summarized in this section also contain results from some smaller ion generators.

### 5.2. Ozone Generation Mechanism and Modelling

The principal mechanism of ozone generation from ESPs has been discussed in previous research [46,48]. In summary, ozone formation in corona discharge can be described according to the following reactions:

$$O_2 + e \rightarrow 2O + e \tag{2}$$

$$O_2 + O + B \rightarrow O_3 + B \tag{3}$$

where *B* represents any species that play a role in supplying or removing kinetic energy from the reaction. On the other hand, ozone is a highly reactive molecule and tends to dissociate. In dry air, ozone dissociates according to the following reactions:

$$O + O_3 \rightarrow 2O_2 \tag{4}$$

$$O_3 + e \rightarrow O + O_2 + e \tag{5}$$

In the presence of water vapor, another possible ozone dissociation mechanism is:

$$O + H_2O \rightarrow 2OH^* \tag{6}$$

$$OH^* + O_3 \rightarrow HO_2 + O \tag{7}$$

$$HO_2 + O_3 \rightarrow OH^* + 2O_2 \tag{8}$$

where *OH\** is a highly unstable intermediate species.

Based on the above ozone formation and dissociation mechanism, Viner et al. [47] developed a simple empirical model to predict the ozone generation rate from ESPs:

$$r_{ozone} = k_1 I - k_2 [O_3][RH] \tag{9}$$

where $k_1$ and $k_2$ are empirically derived constants representing the ozone generation per unit current and the ozone destruction due to water vapor presence, respectively. *I* is the electrode current. $[O_3]$ and [RH] are concentrations of ozone and the relative humidity of the air, respectively.

More complex models are also available. For example, Boelter and Davidson [46] developed an empirical multivariable linear regression model for a two-stage commercial ESP based on experimental data. Theoretical numerical models capable of predicting the ozone generation rate and the spatial distribution of ozone concentration for positive and negative corona discharge wire in dry air have also been developed [49,50].

*5.3. Factors Affecting the Ozone Generation Rate*

There are mechanisms by which ozone generated in ESPs can be affected by both product design and operating conditions.

Product design factors that influence ozone generation include corona type and polarity, current density, applied voltage, discharge electrode/wire diameter, wire material, and the overall geometry of the air cleaner [46,47,51–55]. As discussed in Section 1, ESPs are typically a two-stage operation consisting of a charging (ionizing) and a collecting stage. Ozone formation mainly occurs in the charging (ionization) section. Corona type and polarity is perhaps the most significant factor that affects ozone generation. Positive polarity corona discharges generally generate significantly less ozone compared to negative polarity coronas.

Boelter and Davidson [46] observed that the ozone generated from a tested air cleaner with negative polarity was one order of magnitude greater than that generated with positive polarity. The authors concluded that the most significant differences in positive and negative corona plasma are the size of the plasma region, the distribution of the number density of electrons, and the effect of the gas temperature on that distribution. The current density, which is further determined by the applied voltage (or power) and electrode spacing, also greatly affects ozone generation [46,47,51,52,55]. A linear increase in ozone generation with the increase in current density has been commonly reported, although the maximum current to which the linear ozone–current relationship holds true may vary [46,47,52]. The discharge wire diameter has a moderate effect on ozone generation. A larger wire diameter often leads to higher ozone generation. For example, it has been observed that the 0.10 mm diameter ionizer

wires produced approximately 40% and 30% less ozone than the 0.20mm diameter wires for positive and negative polarity corona, respectively [46,50]. The effect of wire material on ozone generation appeared to be small. The enthalpy of the formation of an oxide coating per oxygen atom, $|-\Delta H_O|$, was found to be a good predictor of the relative effect of wire material for both positive and negative polarities [46]. The lower the $|-\Delta H_O|$, the lower the ozone generation rate. The order of the ozone generation rate with thin metal wires is Ti > Ta > Mo > Ni > Cu > Pd > Ag. Besides thin metal wires commonly used in traditional ESP, carbon nanotubes and carbon fiber ionizers have been used as a corona discharge device in recent studies [51,54,56]. Lower ozone generation rates have been observed in these novel prototypes and such a reduction has been attributed to the decrease in discharge electrode diameter. These results suggest that it is possible to reduce ozone generations from ESPs, which emit elevated levels of ozone to meet relevant standard requirements through proper design improvements.

Operation conditions that affect the ozone generation/concentration include relative humidity (RH), wire temperature, wire/electrode and plate contaminations, air velocity or flow rate, and operating level setting [38,46–48,55,57].

Theoretically, the increase in RH may reduce ozone generation due to ozone dissociation in the presence of water vapor, and the increase in gas temperature may reduce ozone generation through its effects on thermodynamic and transport properties and chemical reaction rates [50]. However, available experimental data are limited and inconsistent. Viner et al. tested several ESPs and observed that the increase in RH modestly increased the ozone generation for negative polarity, while the RH effect could be ignored for positive polarity [47]. Tanasomwang and Lai found an increase in ozone generation at higher RH when testing the long-term performance of two ESPs [48]. Boelter and Davidson tested a two-stage ESP for temperatures ranging from 19 to 28 °C and RH ranging from 7 to 72% [46]. They concluded that changes in RH and air temperature over the ranges expected in homes do not strongly affect ozone generation [46].

More recently, Morrison et al. tested a commercial ducted ESP for temperature ranging from 31 to 41 °C and RH ranging from 30 to 71% [38]. The tested single-stage ESP exhibited higher ozone generation rates at a lower temperature, while no clear trend was observed for varying RH. For the effect of device contamination on ozone generation, the available literature is limited, out of date, or not conclusive. Dorsey and Davidson observed a seven-fold ozone concentration increase after a 7-day operation for a laboratory assembled ESP tested in a wind tunnel using Arizona Road Dust loading, and a 14-times increase after a 42-day operation for a positive polarity two-stage commercial ESP tested using normal unfiltered room air [57]. Tanasomwang and Lai also found increased ozone concentration after a 7-day operation in a restaurant, where cooking products and oil mist may heavily contaminate the electrode surface [48].

However, in their study, the ozone concentration after two weeks of operation showed a reduction over time instead of a further increase. More research is needed to better quantify the impact of soiling and wire or plate contamination overtime on ozone generation for ESPs currently available on the market. As for the airflow rate, studies found that the ozone generation rates remained approximately the same regardless of the airflow rate, although the downstream ozone concentration decreased as the airflow rate (or air velocity) increased [46–48]. Finally, the operating level setting can also affect ozone generation. For example, Jakober and Phillips observed that ozone generation rates at a high setting operation could become 50% higher than the corresponding low setting operation [58].

### 5.4. Range of Measured Ozone Generation and Its Implication on IAQ

The ozone generation from ESP is usually quantified by one of the following three methods [58]: (1) the face test method, in which ozone concentration is measured near the exterior exhaust face of the device; (2) the single-pass test method, similar to the face test method, in which the ozone concentration increase across the ESP is measured using inert ductwork attached to the unit and the ozone generation rate is then directly calculated as the product of ozone concentration increase and airflow rate; and (3) the chamber test method, in which the ozone generation rate is calculated using a

mass-balance model based on the measured average chamber/room/house ozone concentration and ozone natural deposition rate.

Ozone generation rates from ESPs have been measured in several studies using one or more of the above methods and the reported ozone generation rates range from below the detection limit to up to 162 mg/h [28,33,38,46,47,58–60]: Jakober and Phillips tested five portable ionizers (including ESPs) and reported an ozone generation range of 1.3–2.9 mg/h [58]. Niu et al. tested 27 portable ionization-based air cleaners and found that 5 of them produced ozone with a generation rate ranging from 0.06 to 2.8 mg/h [59]. Waring et al. tested three portable ionizers (including one ESP and two smaller ion generators) and observed an ozone generation rate ranging from 3.3 to 4.3 mg/h [33]. Chen et al. measured 1.7 mg/h for a single portable ESP tested in their study [28]. Viner et al. conducted experiments using one table-top and two in-duct ESPs and observed an ozone generation rate of ~2.2 mg/h for the table-top unit and approximately 20–30 mg/h for the in-duct ESPs, respectively [47].

Boelter and Davidson studied the impact of product design and operating parameters on the ozone generation rate using one ESP unit. Depending on the test conditions, the ozone generation rate could reach a maximum of about 20 mg/h (0.0055 mg/s) for positive polarity and as high as 162 mg/h (0.045 mg/s) for negative polarity [46]. More recently, Poppendieck et al. tested two in-duct ESPs in a manufactured test house. They observed that the use of ESPs raised indoor ozone concentration by 20 to 77 ppb and reported an estimated ozone generation rate ranging from 22 to 60 mg/h [60]. Morrison et al. tested two in-duct ESPs in both lab and field test houses. The ozone generation rate was below the level of detection for one ESP and about 21–43 mg/h for the other [38].

Morrison et al. mentioned that outdoor air as a source of ozone can rise to ~100 mg/h for a residence on a highly polluted day [38,61]. The above range of ozone generation rates from ESPs (0–162 mg/h) implies that ESPs, at least some poorly designed devices, have the potential to increase indoor exposure in buildings significantly. To more precisely predict the impact of the ozone generation rate of ESPs on indoor concentration, both mass balance models for a steady-state single zone and a more sophisticated dynamic multi-zone model have been developed [38]. For example, Morrison et al. conducted steady-state simulations for California homes [38].

They defined a Standard Home based on California average values of building volumes, areas, air exchange rates, ozone penetration, and ozone decay rates, and an At-risk House based on a reasonable choice of parameters that, when combined, maximized the resulting indoor ozone concentration. Their results indicated that the increment in indoor ozone concentration could reach 50 ppb, a concentration limit used in both UL Standard 867 and CSA Standard 187 [35,36], when the emission rate from the in-duct ESP is about 150 mg/h for a Standard Home. The same concentration could be reached for an emission rate of only about 27 mg/h for an At-risk House. Morrison et al. [61] also conducted a Monte Carlo analysis of indoor ozone levels in four U.S. cities with the aim of providing guidance to regulatory agencies on setting maximum ozone generation rates from consumer appliances. They found that it was difficult to protect 80% or more of a building's occupied hours from experiencing ozone concentrations greater than 20 ppb in some cities due to outdoor air exchange even with a zero-emission rate and therefore, suggested the need for a zero-emission rate standard from consumer appliances.

In summary, these results suggest that ozone generation from ESPs can be significant. Therefore, standardized ozone generation testing should be conducted for ESPs (including both portable and in-duct) to ensure that they do not cause indoor ozone concentrations to exceed health guidelines.

### 5.5. Other By-Products

Very limited data are available on the generation of by-products by ESPs other than ozone and they mainly focus on fine and ultrafine particles. It is widely established that ozone reactions with indoor unsaturated organic compounds (i.e., terpenoids and terpenes) released by air cleaners, air fresheners, and personal care products can generate respiratory irritants and form a secondary organic aerosol

(SOA) in the ultrafine and fine range [33,62]. Because ESPs can generate ozone as a by-product, it is theoretically possible that the operation of ESP leads to a net increase in indoor particle concentrations.

We suggested that these nanoparticles were directly generated by the discharge wire instead of secondary emissions due to ozone reactions and sputtering on the corona discharge appeared to be the key mechanism. Nanoparticles may also be generated by the use of carbon nanotube (CNT) discharge devices in ESPs and the long-term stability of such a novel material deserves further investigations [54].

## 6. Applications

### 6.1. Portable System

Portable air cleaning systems have been popular equipment for reducing indoor pollutant concentrations in homes [12] and offices [63]; they contained ESPs as well.

It is common for manufacturers of air cleaners to claim that their technologies can remove particles effectively. However, some studies have revealed that manufacturers' claims are not valid, and some of the technologies themselves can cause the generation of ultrafine particles [33,64,65].

In order to determine the capability of an air cleaner for removing pollutants, researchers have introduced three measures: efficiency, CADR, and effectiveness. The efficiency is the fraction of particles that is removed in a single pass through the system and it is presented in percent [66]. Another measure is the CADR, which takes into account the airflow rate through the air cleaner and the particle removal efficiency and is presented in cubic meters per hour ($m^3$/h) [67]. The effectiveness shows how an air cleaner is effective compared with other ways of particle removal such as ventilation or deposition to surface etc. [68]. The effectiveness of an air cleaner presents the relative ability of an air cleaner to remove pollutants from the indoor air of a room, compared with dilution from fresh air and indoor natural decay. The effectiveness is dimensionless.

### 6.2. Short-Term Studies of ESPs in Chambers

Short-term studies in this paper mean that the measurement lasts for less than one week. Kinzer and Moreno [69] measured the efficiency of ESPs in a chamber. The authors reported that ESPs were able to achieve more than 50% efficiency for UFPs (ultrafine particles). Ardkapan et al. [70] evaluated five portable air cleaning technologies, including an ESP with an airflow rate of 300 $m^3$/h in order to determine the effectiveness of the cleaners in removing UFPs. Measurements were carried out in a test chamber. The authors reported that the effectiveness of the ESP to remove UFPs was 0.38. Zuraimi et al. [27] examined 12 different air cleaning technologies, including an ESP with an airflow rate of 800 $m^3$/h in order to determine the effectiveness of the cleaners in removing UFPs. The authors found that the effectiveness of the ESP to remove UFPs was 95%. Morawska et al. [71] studied the performance of a two-stage ESP filter in an ASHRAE test rig in order to determine the efficiency of particles ranging from 0.018 to 1.2 μm. The authors reported single-pass efficiencies ranging from 60% to 98% for particles smaller than 0.1 microns, with lower efficiencies noted at high face velocities. Waring et al. [33] conducted a performance examination of an ESP in a 14.75 $m^3$ stainless steel chamber in order to determine the efficiency for the particle diameter range of 12.6–514 nm. The authors reported that the ESP efficiency is approximately 60% for particle sizes of 0.2 μm and starts to increase slightly above 0.2 μm. Niu et al. [59] tested 27 commercially available portable air cleaners, including ESPs in Hong Kong in a test chamber. Testing demonstrated that the ESP air cleaners emit ozone, and that the emission rates show no correlation with the particulate removal capacity in terms of CADR. Shaughnessy et al. [67,72] have ranked the filters based on a series of measurements in a test chamber. The authors reported that the rank order was ESP >HEPA >extended surface (electrets) >ionizers >ozone generators.

### 6.3. Short-Term Studies of ESPs in Homes

A study with ESP presented by King [73] showed that the particle concentrations were reduced from approx. 28,000 to 960 particles/m$^3$, i.e., a 97% reduction in particles. He also reported decreases in the ESP's efficiency over 4 days. Hart et al. [74] conducted a study in order to evaluate the effectiveness of an ESP in a home where a wood stove was the sole heat source and in a home where a wood stove was used as a supplementary heat source. Particle count concentrations in six particle sizes and particle mass concentrations in two particle sizes were measured for 10 12-hour purifier-on and 10 purifier-off trials in each home. Particle count concentrations were reduced by 61–85%. Weichenthal et al. [75] examined a crossover, including 37 residents in 20 homes. Each home received an electrostatic air filter and a placebo filter for one week in random order. PM$_{2.5}$ was monitored throughout the study period. The authors reported that average indoor concentrations of PM2.5 decreased substantially during air filter weeks relative to placebo. The difference between filter and non-filter periods was 37 μg/m$^3$.

### 6.4. Short-Term Studies of ESPs in Offices

Shaughnessy et al. [72] tested an ESP in office rooms with smoking. He reported that the CADR was reduced by 38% for the ESP. Skulberg et al. [63] conducted an intervention study in six office buildings in order to investigate the effect of ESPs on airborne dust and the health of employees. The ESP used in this intervention study had an electrostatic potential of 5000 V with an airflow of 300–470 m$^3$/h. The experiment was performed during two successive periods of 3 weeks. The authors reported that the installation of ESPs reduced the total airborne dust concentration in offices by 46%. Reduction was observed for all particle sizes. Ardkapan et al. [76] evaluated five portable air cleaning technologies, including an ESP with an airflow rate of 300 m$^3$/h to determine the effectiveness of the cleaners in removing UFPs. Measurements were carried out in an office room. The authors reported that the effectiveness of the ESP to remove UFPs was 0.68.

### 6.5. Short-Term Studies of ESPs in Schools

Wargocki et al. [77] conducted an intervention study in five public elementary schools to determine whether reducing the concentration of airborne particles in school classrooms improves school children's performance of their homework and whether the condition of the bag filter in the ventilation system affects this. The authors concluded that the electrostatic air cleaners considerably reduced the concentration of particles in the classrooms. The lower the outdoor air supply rate, the greater the effect. Mattsson and Hygge [78] studied the possible benefits of air cleaners for children with allergies or hypersensitivity in two pairs of classrooms. The study was performed during the pollen season. The authors concluded that air cleaners reduced the concentration of airborne particles and tended to reduce the amount of cat pollen.

### 6.6. Long-Term Studies of ESPs in Homes

Long-term studies in this paper mean that the measurement lasted more than one week. Wallace et al. [79] compared the removal rates of UFPs in cases when no fans were in use, fans were in use, and mechanical filtration or ESPs were employed. All measurements were made over a 2-year period in an occupied townhouse. During the experimental period, the house was occupied by two non-smoking adults. Two types of in-duct filters were tested, including an ESP. The ESP positively charged particles with ionizing wires at 6200 V. The efficiency of the ESP varied considerably depending on the time since the last cleaning of the wires and plates. Efficiencies exceeded 90% for fine particles and 99% for coarse particles soon after cleaning, but began dropping below 90% after several hundred hours of operation for the fine particles and after a few hundred hours more for the coarse particles.

### 6.7. Positioning of a Portable Air Cleaner

Laboratory tests of portable air cleaners are typically conducted in test chambers with ventilation conditions close to complete mixing. The impacts of mixing condition, the relative location of a portable air cleaner, the particle source, and also its effectiveness have been investigated in several studies. For example, Novoselac and Siegel [13], conducting both measurements and CFD simulations, concluded that the assumption of complete mixing within an entire apartment can overestimate the particle exposure reduction by as much as a factor of 2. Effective positioning of a portable air cleaner (a good choice of location) can result in a change by a factor of 2.5 in overall particle removal. Consequently, the positioning of the unit may strongly influence occupants' particle exposure. However, a reasonable strategy would be to position the unit in the central area of the residence and/or close to the particle source. The CADR must be sufficiently high and thus, the air cleaner must be chosen with respect to the magnitude of other particle removal mechanisms, e.g., the ventilation rate.

### 6.8. In-Duct System

ESP air cleaners are applied in HVAC systems of residential and commercial buildings and spaces as well as in vehicles such as cars, buses, or rail. The primary function of these applications is to remove particulate matter from handled air [80,81]. ESPs are part of a larger system and thus, their design and operation must match, e.g., fan operation, speed, location of humidifier, outdoor air intake, evaporator, etc.

In-duct ESPs are typical of two-stage operations, i.e., ionization and collector stage. In the first stage, corona discharge ionizes flyby particles, while in the second stage, particles are collected on grounded surfaces. The ionizing components are mostly wire or spike and plate, or pin and perforated plate, but also a wire or spike in a grounded cylinder. In HVAC applications, the ionizing electrode is typically spaced by tenths of millimeters from a grounded surface/electrode. A positive voltage is connected to a wire, spike, or needle while the plate is grounded. This arrangement creates less corona/ionization, generates less ozone, and thus, is more suitable for use in occupied spaces.

Specific developments were published in terms of material selection of the wire (tungsten, carbon), ionization electrode shape and thickness [82–84], or power control, e.g., continuous direct current, alternating current, or high frequency charging [80]. The collector can be arranged in a parallel plate configuration where the grounded plate alternates with a plate at high voltage or as a honeycomb. It can be made solely of metal or from metal insulated by electric-resistant non-metallics/plastics [85–87]. The distance between collector electrodes could be as small as a few millimeters up to several millimeters. Several studies researched the impact of the curvature waves or grooves of the grounded plates at the ionizer stage and the collector stage [81,88].

High voltage metal collecting plates require safety shielding. The user needs to be protected from the possibility of electrical shock from the high voltage power supply. Even when the power supply is switched off, there is a danger of shock from the stored electrical charge on the plates. The plates need to be removed for cleaning and therefore, a safety interlock is usually provided for automatic discharge from plates before gaining access to them. Therefore, a collector type with a plastic coating was developed to reduce the risk of arcing and enable high electrostatic strength—higher filtration efficiency. The plastic material may be of polypropylene, polyethylene, or copolymer, but also PVC, PET, PTFE, or polycarbonate. The design of such an ESP could, for example, be a multilayered "fluted" monolithic block of polypropylene. The filter block is constructed by stacking layers of corrugated polypropylene interwoven with thin conductive electrodes, with every other electrode biased to a high voltage.

Zuraimi and Tham [89] evaluated the impact of in-duct filters operating in an office building. The performance of media filters (grade M5 by EN 779), electrostatic precipitation filters, and electrostatic precipitation filters enhanced with a media pre-filter (grade G1 by EN 779) was compared with regard to the reduction in particles ranging from 0.3 to about 5 micrometers. The ESP filter was a two-stage type with ionizing wires and collecting plates of voltage about 8 and 4 kV, respectively. While the new

media filter was 30% efficient regarding the removal of fine particles, the ESP was 80% efficient and the ESP with pre-filter filtration efficiency was close to 100%. They further emphasize the importance of a pre-filter, which together with the ESP may perform as well as HEPA, but without the high pressure drop. They thus suggest that improved efficiencies are due to the backpressure resulting from the face of the pre-filter (obstruction caused by the pressure drop) and not by the combined filtration efficiencies of the two types of filters.

Croxford et al. [90] found that ESPs used in offices reduced airborne dust significantly. They reduced the dust level in the breathing zone, while finding no difference in surface dust levels. Furthermore, the authors concluded that ESPs might be highly efficient at removing particles with a small aerodynamic diameter.

Skulberg et al. [63] conducted an intervention study to identify health improvements in the upper and lower airways of office workers after the installation of local electrostatic air cleaners. Its electrostatic potential was 5 kV. The authors utilized subjective symptoms using a questionnaire and indexes calculated for general, irritation, and skin symptoms. Objective respiratory health indicators were recorded with acoustic rhinometry and peak expiratory flow (PEF) meters. In the intervention group, there was a decrease in mean dust concentration from 65 to 35 $\mu$g/m$^3$ and a reduction from 57 to 47 $\mu$g/m$^3$ in the control group ($p < 0.05$ for difference in decline). A reduction was observed for all particle sizes. The irritation and general symptom indices decreased in both groups, but there was no improvement in the intervention group compared to the control group.

Wargocki et al. [77] studied the performance of students in classrooms with and without in-duct electrostatic precipitators. Operating the electrostatic air cleaners considerably reduced the concentration of particles ranging from 20 nanometers to particles larger than 15 micrometers in the classrooms. The effect was greater the lower the outdoor air supply rate.

Morawska et al. [71] studied the effect of face velocity and the nature of aerosol on the collection of sub-micrometer particles by the ESP. The authors reported that an electrostatic air cleaner showed a minimum - fractional removal efficiency in the particle size interval 0.1–0.45 $\mu$m, and also that the efficiency dropped for particles smaller than 20 nm. However, the removal efficiency appeared to be practically independent of the particle size when the air velocity through the air cleaner was low. At an airflow rate of 472 L/s, the efficiency for 0.2 $\mu$m particles was about 95%. When the airflow rate was increased slightly, i.e., more than twofold, to 1050 L/s, the efficiency dropped to about 55%.

## 7. Conclusions and Recommendations

The following conclusions can be drawn regarding the testing and performance of electrostatic precipitators:

- Both in-duct and portable air cleaners have advantages and disadvantages; in-duct units purify all the air entering the duct and consequently, distribute clean air to every room, through the supply devices; portable units only purify the air in the room in which they are placed, but have the advantage of reducing the risk due to cross contamination between rooms. Another minor advantage of in-duct units is no noise in comparison with portable units which have low levels of noise.

- Ozone is the major by-product of ESPs. The mechanisms suggest that the ozone generation rate can be affected by both product design and operating conditions. Product design factors that influence ozone generation include corona type and polarity, current density, discharge electrode/wire diameter, wire material, and the overall geometry of the air cleaner. Operating conditions that influence ozone generation/concentration include relative humidity, temperature, wire/electrode (and plate) contaminations, and the airflow rate. The reported ozone generation rates from ESPs range from below the detection limit to up to 162 mg/h. Standardized ozone generation testing is needed to ensure that they do not cause indoor ozone concentration to exceed the health guidelines.

- None of the available standards consider performance with respect to ultrafine particles.

- All standards focus on the determination of the performance of new, unused air cleaners. No test standards address the potential generation of by-products other than ozone.
- ESPs have the lowest noise generation of all tested units equipped with a fan, and in addition, ESPs use less power than other units.
- ESPs have a lower pressure drop compared to mechanical filters with comparable particle removal efficiencies.
- The positioning of a portable air cleaner also affects the overall particle removal and consequently, influences occupants' exposure to particles.
- In HVAC applications, the ionizing electrode is typically spaced by tenths of millimeters from a grounded surface/electrode. Positive voltage is connected to a wire, spike, or needle, while the plate is grounded. This arrangement creates less corona/ionization, generates less ozone, and is thus more suitable for use in occupied spaces.

**Author Contributions:** A.A. was the editor and contributed to Sections 1 and 6. S.R. contributed to Section 6. W.C. wrote the initial draft of Section 5. She also contributed to the overall review and editing of the paper. J.M. wrote the initial draft of Sections 2 and 5. He also contributed to the overall review and editing of the paper. L.F. contributed to Section 6. S.Z. contributed to Section 3. L.E. contributed to Section 4. J.Z. contributed to Section 3 and reviewed the manuscript. J.S. and P.W. reviewed the manuscript. All authors contributed to Section 7. All authors have read and agreed to the published version of the manuscript.

**Funding:** This research was funded by Aalborg University and National Natural Science Foundation of China (grant numbers: 52078269, 51722807).

**Acknowledgments:** This research is a joint effort through the International Society of Indoor Air Quality and Climate (ISIAQ) STC 22 on Air cleaning.

**Conflicts of Interest:** The authors declare no conflict of interest. The funders had no role in the design of the study; in the collection, analyses, or interpretation of data; in the writing of the manuscript, or in the decision to publish the results.

**Disclaimer:** Conclusions and opinions are those of the individual authors and do not necessarily reflect the policies or official views of the California Department of Public Health.

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
