# Peer review of "Electrostatic Precipitators as an Indoor Air Cleaner—A Literature Review"

_sustainability, doi:10.3390/su12218774_

Round 1

Reviewer 1 Report

An interesting paper giving a broad literature review background of ESP systems. The paper could be strengthened if the discussion in some sections (such as on the by-products in section 5) was to include more quantitative performance factors for the equipment reviewed.  References are given to the original papers, but it would be nice to see quantitative values such as air flow through the system or wattage on power settings, as opposed to “Hi” or “Low” settings.

P.3, Figure 2:  Is there further information that can be given as to the length of the “Active Zone” vs. “Passive Zone”.

P.5, Figure 3:  Does this figure apply to both single- and two-stage preceptors?  This are two descriptions on the prior page, but the there is only one design presented in this figure.

P.2, line 55-56:  It is said “The energy used for air conditioning of buildings makes up almost 40% of the total energy consumption.”

It would be appropriate to provide a reference for this number.

P.2, line 78:  It is said “_ENREF_15were …”

There should be a space between these words.

P.3, line 105:  The link to the reference is missing.

P.3, line 105-106:  It is said “… e lectrons …”

This word should not be split between lines.

P.4, line 145:  The link to the reference is missing.

P.4, line 145-146:  It is said “… a re …”

This word should not be split between lines.

P.9, line 295:  “Where, …” should not be capitalized.

P.9, line 321:  “Where, …” should not be capitalized.

P.10, paragraph starting line 346:  Was a reason given as to why larger vs. smaller diameter wires performed differently? 

P.10, line 375:  The wrong symbol is used for degree in “28oC” 

P.10, line 379 to page 11 line 421:  The font has changed for some reason.

Reviewer 2 Report

Review: "Electrostatic Precipitators as an Indoor Air Cleaner- A Literature Reivew."

This paper reviewed the electrostatic precipitate system used in indoor air purifiers, and investigated the principle, type, performance test method, particle removal performance, by-products, and application examples (schools, houses, buildings) of electrostatic precipitators.

In the paper, the indoor electrostatic precipitate system has been confirmed through sufficient references and reviews, so there seems to be no shortage for publication in this journal. However, it can be a better review paper if the technical parts and the commercial product, product specification, design method, etc. for the currently available electrostatic precipitators are added.

In conclusion, the English expression of this paper does not seem to have any major problems, and it is considered that the direction of the review and various viewpoints are applied to be published in the journal of Sustainablity.

Reviewer 3 Report

The scope of this study is admirable and has required a multi and interdisciplinary team to generate and test a range of techniques to improve indoor air quality. The conclusions have generated many new hypothesis to test, as well as  highlighting the possible downsides of some types of electro-static air purifiers. It is a significant piece of work and moves the science forward.

There are some minor typos, missing references and at one point a strange change in text size that will need to be corrected. 
